# Critical enzymes for biosynthesis of cucurbitacin derivatives in watermelon and their biological significance

Young-Cheon Kim[1], Daeun Choi[1], Ahra Cha[1], Yeong-Geun Lee [2], Nam-In Baek[2], Suman Rimal[3], Jiun Sang[3], Youngseok Lee [3] & Sanghyeob Lee [1,4✉]

Various cucurbitacins have been isolated, and their structures have been elucidated. Owing to their economic potential and importance as active pharmacological compounds, their cytotoxicity in various cancer cells has been assessed. Here, we mined several candidate genes with potential involvement in cucurbitacin biosynthesis in watermelon (*Citrullus lanatus*) and performed in vitro enzymatic assays and instrumental analyses using various substrates to identify cucurbitacin functions and products. Enzymatic activities of two acetyltransferases (ACTs) and one UDP-glucosyltransferase (UGT) against cucurbitacins were confirmed, resulting in the synthesis of novel cucurbitacins in vivo and/or in vitro to our knowledge. As ACTs and UGT are involved in the dynamic conversion of cucurbitacins by catalyzing acetylation and glucosylation at moieties in the cucurbitacins skeleton, these findings improve our knowledge on how these genes contribute to the diversity of cucurbitacins.

[1] Department of Bioindustry and Bioresource Engineering, Sejong University, Seoul 05006, Korea. [2] Department of Oriental Medicinal Biotechnology, Kyung Hee University, Gyeonggi-do 17104, Korea. [3] Department of Bio and Fermentation Convergence Technology, BK21PLUS Project, Kookmin University, Seoul 02707, Korea. [4] Plant Engineering Research Institute, Sejong University, Seoul 05006, Korea. ✉email: sanglee@sejong.ac.kr

Plants produce diverse natural compounds that have ecologically important functions. A few of these natural compounds participate directly in growth and development, acting as growth regulators, photosynthetic pigments, and antioxidants[1]. Most natural compounds (phytochemicals), or secondary metabolites, perform specialized functions against abiotic and biotic stress, and provide beneficial interactions and communication with other plant/non-plant species[2,3]. To date, several phytochemicals have been produced commercially by in vitro tissue culturing systems, which culture plant organs, explants, specific tissue, and protoplast[4,5].

Phytochemicals are grouped into three major classes: terpenes, alkaloids, and phenylalanine. Terpenes, such as di-, tri-, tetra, and polyterpenes, are non-volatile and are generally believed to possess biological functions[6,7]. Cucurbitacins are structurally diverse triterpenoids originally found in *Cucurbitaceae* species and possess enormous pharmacological potential[8]. Cucurbitacins and their derivatives can be categorized into 12 main groups according to variations in their side-chains[9]. Cucurbitacins A, B, C, E, I, J, and K and their glycosides are abundant in *Cucurbitaceae* family members; they induce a cytotoxic activity that inhibits cancer cell proliferation, actin polymerization, capillary permeability, and anti-inflammatory activity[10–13]. However, published data that provide empirical knowledge on the function of cucurbitacins within plants are limited. Cucurbitacins are synthesized via lanosterol, cycloartenol, or parkeol because the tetracyclic cucurbitacin skeleton shares high similarity with that of steroidgenic triterpenes[14,15]. Cucurbitacin synthesis begins with the cyclization of 2,3-oxidosqualene to cucurbitadienol by oxidosqualene cyclase (OSC), and further metabolizations occur by subsequent hydroxylation, acetylation, and glucosylation steps, producing a variety of cucurbitacins[9]. Some enzymes, including OSC, cytochrome P450 (CYP), acetyltransferase (ACT), and UDP-glucosyltransferase (UGT), which are potentially involved in the biosynthesis of cucurbitacins, have been isolated and characterized from pumpkins (*Cucurbita pepo*), watermelons (*Citrullus lanatus*), cucumbers (*Cucumis sativus*), and melons (*C. melo*) using comparative genomic analysis[14,16–18]. However, information on gene and enzyme functions involved in cucurbitacin core skeleton biosynthesis and their modifications to produce various cucurbitacins is lacking. In a previous study, we investigated the composition and levels of cucurbitacins in different tissues of *C. colocynthis* and *C. lanatus* germplasms using liquid chromatography-mass spectrometry (LC-MS) and high-performance liquid chromatography (HPLC) and identified a correlation between bitterness and cucurbitacin concentration[19]. More diverse compositions and higher concentrations of cucurbitacins were detected in *C. colocynthis* than in *C. lanatus*, and the extent of bitterness corresponded with cucurbitacin content.

The present study was designed to identify enzyme-coding genes that enhance the diversity of cucurbitacins and to investigate the potential functions of newly identified cucurbitacins. Here, we show that some enzymes act at important functional steps in the biosynthetic pathway, generating diverse cucurbitacins in *Cucurbitaceae* plants and introducing few novel cucurbitacins in *Citrullus* species to our knowledge.

## Results

**ACT3 catalyzes the C16-acetylation specificity.** Cucurbitacins are a multiplex category of diverse compounds in the Cucurbitaceae family[8]. Cucurbitacins are known for their bitter taste, which is derived from cucurbit, and to date, 12 categories of cucurbitacins have been identified[20]. Diversity in cucurbitacins results from the variety of side-chain derivatives, which influence their biological activity[8,21,22]. Several reports have shown that

Cucurbitacin biosynthetic pathways converge in cucumber, melon, and watermelon[16–18]. However, limited information is available to explain the formation of divergent cucurbitacin derivatives. Here, we used RNA-seq and in vitro enzyme assays to investigate the importance of newly identified *ACT* and *UGT* genes for the biosynthesis of divergent cucurbitacin derivatives.

Previously, we showed that the content of cucurbitacins is higher in PI532627 (USDA plant ID) watermelon seeding plants (*C. lanatus*) than in PI536451 (USDA plant ID). To survey genes related to cucurbitacin-derivative biosynthesis pathways, we performed RNA-seq analysis using these two watermelons germplasm (PI532627 and PI536451), which have different cucurbitacin profiles[19]. In total, 54,301,266 and 52,981,816 reads were acquired from PI532627 and PI536451, respectively (Supplementary Table 1). Since CYP450s, ACTs, and UGTs are involved in cucurbitacin biosynthesis pathways, 16 genes tentatively annotated as gene-coding for CYP450s, ACTs, and UGTs were selected. We found increased transcript levels, especially in cucurbitacin E- (CuE) and cucurbitacin E-2-O-glucoside (CuE-Glu)-rich seedlings (Supplementary Table 2). Real-time quantitative polymerase chain reaction (RT-qPCR) confirmed the reliability of RNA-seq data (Supplementary Fig. 1).

Yeast engineering can help determine the role of CYP450s in different parts of the cucurbitacin biosynthesis pathway. On this line, yeast harboring multiple genes are developed. Moreover, mass yeast culturing and high-quality purification systems are needed for structural analysis. This type of experimental approach requires multi-combination constructions of yeast expression vectors, which can be constructed in a timely manner. However, we aimed to elucidate the function of ACTs and UGTs in cucurbitacin biosynthesis. ACTs and UGT enzymatic studies were conducted using major cucurbitacins, which are commercially available, as substrates. Unfortunately, only one UGT74F2 (Cla004392) soluble protein was successfully purified out of seven UGTs. Additionally, three soluble ACT (ACT1 [Cla007081], ACT2 [Cla008353], and ACT3 [Cla022713]) proteins were purified in our recombinant protein expression system for use in in vitro enzyme activity assays. Previous studies on cucumber, melon, and watermelon showed that *ACTs* (*Csa6G088700, Melo3C022373*, and *Cla007081*) specifically acetylated the C25-hydroxyl of cucurbitacin D (CuD), cucurbitacin I (CuI), and deacetyl-cucurbitacin C in the synthesis of cucurbitacin B (CuB), CuE, and cucurbitacin C (CuC), respectively[16,18]. However, except for C25-hydroxyl, these ACTs were unable to acetylate other hydroxyl moieties of cucurbitacin[18]. Cucumber UGT73AM3 (Csa3G744990), a UGT enzyme cloned from cucumber plant, presented substrate specificity toward CuC[17]. These results indicate a missing link between *ACT* and *UGT* genes, which is required to elucidate side-chain modifications, leading to the formation of divergent cucurbitacin derivatives. To investigate substrate specificity of the *ACT* and *UGT* genes identified in this study, an enzymatic in vitro assay was conducted using several cucurbitacins as substrates, including CuB, CuD, CuI, CuE, and CuE-Glu. Interestingly, ACT3 could acetylate all analyzed cucurbitacins during synthesis and one acetylated cucurbitacin during LC-MS analysis subsequent to enzymatic reaction (Fig. 1a–e and Supplementary Figs. 2a, 4a, 6a, 8a, 10a). These cucurbitacins were identified as 16-O-acetyl CuB (Supplementary Figs. 2b and 3a, b), 16-O-acetyl CuD (Supplementary Figs. 4b and 5a, b), 16-O-acetyl CuI (Supplementary Figs. 6b and 7a, b), 16-O-acetyl CuE (Supplementary Figs. 8b and 9a, b), and 16-O-acetyl CuE-Glu (Supplementary Figs. 10b and 11a, b) using NMR spectroscopy (Tables 1 and 2). Although 16-O-acetyl CuB, 16-O-acetyl CuE, and 16-O-acetyl CuE-Glu were previously isolated from *Echinocystis esiacea*, *Bacopa monnieri*, and *Gratiola officinalis* respectively[23–25], 16-O-acetyl CuD and 16-O-acetyl CuI

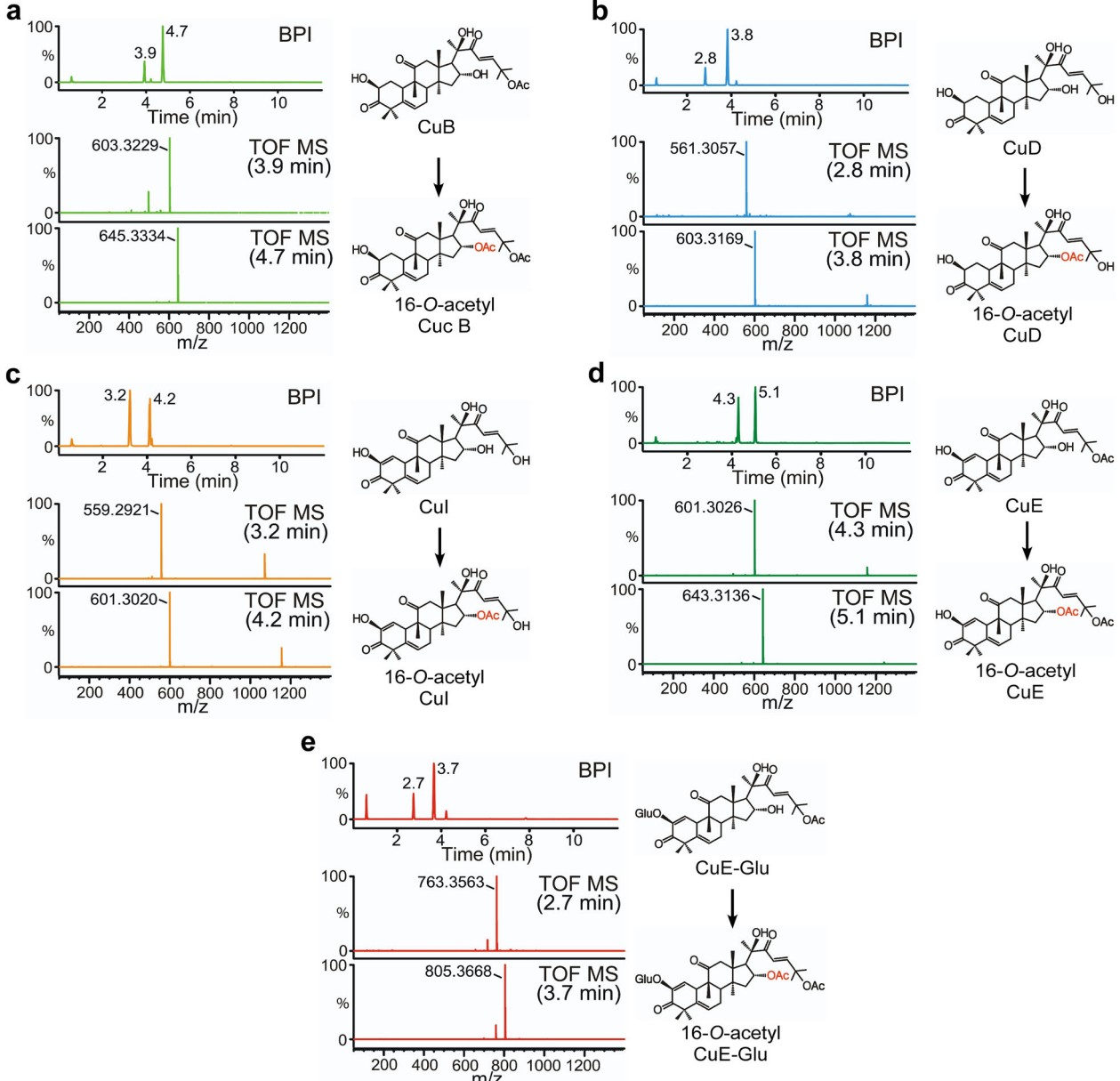

**Fig. 1 ACT3 catalytic activity in the biosynthesis of new compounds. a** LC-MS analysis of ACT3 enzymatic reaction using CuB as a substrate. The base peak intensity (BPI) chromatogram in negative-ion mode shows peaks at retention times (RT) of 3.9 and 4.7 min Peaks in the extracted ion chromatogram at a mass/charge ratio ($m/z$) of 603.3229 [M+formic acid (FA)-H]$^-$ and 645.3334 [M+FA-H]$^-$ correspond to CuB and 16-$O$-acetyl CuB, respectively. **b** LC-MS analysis of ACT3 enzymatic reaction using CuD as a substrate. The BPI chromatogram in negative-ion mode shows peaks at 2.8 and 3.8 min. Peaks in the extracted ion chromatogram at $m/z$ 561.3057 [M+FA-H]$^-$ and 603.3169 [M+FA-H]$^-$ correspond to CuD and 16-$O$-acetyl CuD, respectively. **c** LC-MS analysis of ACT3 enzymatic reaction using CuI as a substrate. The BPI chromatogram in negative-ion mode shows peaks at 3.2 and 4.2 min. Peaks in the extracted ion chromatogram at $m/z$ 559.2921 [M+FA-H]$^-$ and 601.3020 [M+FA-H]$^-$ correspond to CuI and 16-$O$-acetyl CuI, respectively. **d** LC-MS analysis of ACT3 enzymatic reaction using CuE as a substrate. The BPI chromatogram in negative-ion mode shows peaks at 4.3 and 5.1 min. Peaks in the extracted ion chromatogram at $m/z$ 601.3026 [M+FA-H]$^-$ and 643.3136 [M+FA-H]$^-$ correspond to CuE and 16-$O$-acetyl CuE, respectively. **e** LC-MS analysis of ACT3 enzymatic reaction using CuE-Glu as a substrate. The BPI chromatogram in negative-ion mode shows peaks at 2.7 and 3.7 min. Peaks in the extracted ion chromatogram at $m/z$ 763.3563 [M+FA-H]$^-$ and 805.3668 [M+FA-H]$^-$ correspond to CuE-Glu and 16-$O$-acetyl CuE-Glu, respectively. The structure of the product (right) was elucidated using LC-MS and NMR spectroscopy. Error bars denote ± standard deviation (SD), $n = 3$.

represent new compounds that were identified in this study. ACT3 demonstrated substrate promiscuity and specific acetylation activity at the C16-hydroxyl of the cucurbitacin skeleton. We assessed the ability to acetylate the C16-hydroxyl moiety with specificity as well as the basic kinetic parameters to characterize the efficiency and affinity of ACT3 toward cucurbitacins. The $K_m$ values for CuB, CuD, and CuE-Glu were similar; however, the

$k_{cat}$, $k_{cat}/K_m$, and $V_{max}$ values were the highest for CuE-Glu. Interestingly, the $k_{cat}$, $k_{cat}/K_m$, and $V_{max}$ values of ACT3 for CuE and CuE-Glu were significantly different, and this low catalytic efficiency was probably derived from a sugar moiety (Table 3). To confirm whether 16-$O$-acetylated cucurbitacins are also detectable in vivo, watermelon leaves extracts fractionated using HPLC system were analyzed by LC-MS. Most cucurbitacin substrates

**Table 1 The $^1$H-NMR data of compounds 1–5 in CD$_3$OD [$\delta_H$ (600 MHz) in ppm, coupling pattern, $J$ in Hz.**

| No. | Compound 1 | Compound 2 | Compound 3 | Compound 4 | Compound 5 |
|---|---|---|---|---|---|
| 1 | 2.97, m; 1.19, m | 2.97, m; 1.21, m | 5.74, d, $J$ = 3.0 Hz | 5.95, d, $J$ = 3.0 Hz | 6.14, d, $J$ = 2.4 Hz |
| 2 | 4.55, m | 4.56, br. dd, $J$ = 12.8, 5.7 Hz | | | |
| 6 | 5.79, br. d, $J$ = 5.4 Hz | 5.79, br. d, $J$ = 5.4 Hz | 5.78, m | 5.81, m | 5.85, m |
| 7 | 2.41, br. dd, $J$ = 18.0, 5.4 Hz; 1.99, overlapped | 2.41, br. dd, $J$ = 19.8, 5.4 Hz; 1.94, overlapped | 2.38, m; 1.96, m | 2.47, m; 2.02, m | 2.43, m; 2.02, m |
| 8 | 2.01, overlapped | 2.01, br.d, $J$ = 7.4 Hz | 2.08, d, $J$ = 8.4 Hz | 2.10, d, $J$ = 8.4 Hz | 2.13, d, $J$ = 8.4 Hz |
| 10 | 2.97, br. d, $J$ = 12.6 Hz | 2.97, br. d, $J$ = 12.7 Hz | 3.63, br. s | 3.66, br. s | 3.70, br. s |
| 12 | 3.47, d, $J$ = 14.4 Hz; 2.65, d, $J$ = 14.4 Hz | 3.49, d, $J$ = 14.7 Hz; 2.65, d, $J$ = 14.7 Hz | 3.45, d, $J$ = 15.0 Hz; 2.68, d, $J$ = 15.0 Hz | 3.48, d, $J$ = 15.0 Hz; 2.70, d, $J$ = 15.0 Hz | 3.44, d, $J$ = 15.0 Hz; 2.71, d, $J$ = 15.0 Hz |
| 15 | 1.94, dd, $J$ = 14.4, 6.6 Hz; 1.38, br. d, $J$ = 14.4 Hz | 1.94, overlapped; 1.35, br. d, $J$ = 14.7 Hz | 2.03, dd, $J$ = 15.0, 7.8 Hz; 1.39, overlapped | 2.01, dd, $J$ = 15.0, 7.8 Hz; 1.43, overlapped | 2.03, dd, $J$ = 15.0, 7.8 Hz; 1.45, overlapped |
| 16 | 5.37, br. dd, $J$ = 14.4, 6.6 Hz | 5.30, br. dd, $J$ = 14.7, 7.4 Hz | 5.32, br. dd, $J$ = 7.8, 7.8 Hz | 5.40, br. dd, $J$ = 7.8, 7.8 Hz | 5.41, br. dd, $J$ = 7.8, 7.8 Hz |
| 17 | 2.63, d, $J$ = 14.4 Hz | 2.83, d, $J$ = 7.4 Hz | 2.82, d, $J$ = 7.8 Hz | 2.82, d, $J$ = 7.8 Hz | 2.82, d, $J$ = 7.8 Hz |
| 18 | 0.96, s | 0.94, s | 0.98, s | 1.01, s | 1.05, s |
| 19 | 1.05, s | 1.05, s | 0.99, s | 1.01, s | 1.02, s |
| 21 | 1.39, s | 1.41, s | 1.41, s | 1.42, s | 1.42, s |
| 23 | 6.76, d, $J$ = 15.6 Hz | 6.83, d, $J$ = 15.4 Hz | 6.82, d, $J$ = 15.6 Hz | 6.80, d, $J$ = 15.6 Hz | 6.80, d, $J$ = 15.6 Hz |
| 24 | 7.06, d, $J$ = 15.6 Hz | 7.05, d, $J$ = 15.4 Hz | 7.04, d, $J$ = 15.6 Hz | 7.11, d, $J$ = 15.6 Hz | 7.10, d, $J$ = 15.6 Hz |
| 26 | 1.54, s | 1.33, overlapped | 1.33, s | 1.59, s | 1.59, s |
| 27 | 1.56, s | 1.33, overlapped | 1.33, s | 1.58, s | 1.58, s |
| 28 | 1.28, s | 1.28, s | 1.24, s | 1.27, s | 1.29, s |
| 29 | 1.27, s | 1.27, s | 1.28, s | 1.32, s | 1.32, s |
| 30 | 1.33, s | 1.33, overlapped | 1.35, s | 1.38, s | 1.38, s |
| 1′ | | | | | 4.66, d, $J$ = 7.8 Hz |
| 2′ | | | | | 3.42, overlapped |
| 3′ | | | | | 3.43, overlapped |
| 4′ | | | | | 3.54, overlapped |
| 5′ | | | | | 3.63, overlapped |
| 6′ | | | | | 4.02, dd, $J$ = 12.0, 2.4 Hz; 3.88, dd, $J$ = 12.0, 3.6 Hz |
| Ac-Me | 1.88, s | 1.83, s | 1.84, s | 1.90, s | 1.91, s |
| Ac-Me | 2.01, s | | | 2.02, s | 2.03, s |

Compound 1 represent 16-O-acetyl CuB. Compound 2 represent 16-O-acetyl CuD. Compound 3 represent 16-O-acetyl CuI. Compound 4 represent 16-O-acetyl CuE. Compound 5 represent 16-O-acetyl CuE-Glu.

(CuB, CuD, CuI, CuE, and CuE-Glu) and their 16-O-acetylated forms were detected in fractions collected according to their retention time, coinciding with substrate standards and their products (Supplementary Fig. 12a–i). Taken together, these results suggest that *ACT3* (*Cla022713*) may be involved in a cucurbitacin biosynthesis step by encoding an acetyltransferase that catalyzes the conversion of CuB, CuD, CuI, CuE, and CuE-Glu into 16-O-acetylated cucurbitacin forms, contributing to the diversity of cucurbitacins in watermelon plants.

**ACT1 acts as an acetyltransferase and deacetylase.** To investigate the enzymatic activity of ACT1 (Cla007081) and ACT2 (Cla008353) against CuB, CuD, CuE, CuE-Glu, and CuI, in vitro enzymatic assays were conducted and the purified products were analyzed by HPLC. ACT2 presented no activity against the cucurbitacins tested as substrates (Supplementary Fig. 13). Surprisingly, ACT1 catalyzed the deacetylation of CuB, CuE, and CuE-Glu (Supplementary Fig. 14), which is in contrast to previous results demonstrating the production of C25-O-acetylated forms of CuD, CuI, and deacetyl-CuC[16,18]. Additional enzymatic assays were performed to verify whether ACT1 could acetylate or deacetylate C25 on the 16-O-acetyl cucurbitacins (CuB, CuD, CuI, CuE, and CuE-Glu). Since these are not commercially available, in vitro primary enzymatic reactions were conducted with ACT3 using CuB, CuD, CuI, CuE, and CuE-Glu.

Subsequently, purified primary reaction products passed through a secondary in vitro enzymatic process using ACT1. Then, the cucurbitacins purified after the secondary enzymatic process were analyzed by HPLC and/or a LC-MS system. Interestingly, ACT1 acetylated C25 on 16-O-acetyl CuD and 16-O-acetyl CuI in production of 16-O-acetyl CuB and 16-O-acetyl CuE (Fig. 2a, b). Moreover, it deacetylated the same moiety on 16-O-acetyl CuB and 16-O-acetyl CuE precursors resulting in the production of 16-O-acetyl CuD and 16-O-acetyl CuI, respectively (Fig. 2c, d). Although deacetylation of CuE-Glu and 16-O-acetyl CuE-Glu by ACT1 was confirmed by LC-MS analysis (Fig. 2e), acetylation of CuI-Glu and 16-O-acetyl CuI-Glu was not. This was because of the low amount of CuI-Glu available for the enzymatic assay. Our results showed that ACT1 might acetylate and deacetylate C25 cucurbitacins with specificity, and exhibits precursor promiscuity.

**CuE is glucosylated by UDP-glucosyltransferase UGT74F2.** Functional glucosyltransferases are promiscuous enzymes against a wide range of sugar acceptors[26]. To investigate the promiscuous function of UGT74F2, in vitro glucosyltransferase assays were performed using CuB, CuD, CuI, and CuE as sugar acceptors. LC-MS analysis revealed that CuE was glucosylated by UGT74F2 (Fig. 3a). Interestingly, UGT74F2 could not glycosylate the other forms of cucurbitacin, including CuB, CuD, and CuI (Supplementary Fig. 15). This indicates that UGT74F2 may prefer certain

**Table 2 The $^{13}$C-NMR data of compounds 1–5 in CD$_3$OD [$\delta_c$ (150 MHz) in ppm].**

| No. | Compound 1 | Compound 2 | Compound 3 | Compound 4 | Compound 5 |
|---|---|---|---|---|---|
| 1 | 37.2 | 37.3 | 116.7 | 116.7 | 123.5 |
| 2 | 75.4 | 75.5 | 147.0 | 150.0 | 147.4 |
| 3 | 215.2 | 214.0 | 200.1 | 200.0 | 199.8 |
| 4 | 51.7 | 51.6 | 49.0 | 49.2 | 49.7 |
| 5 | 142.2 | 142.3 | 138.6 | 138.6 | 137.6 |
| 6 | 121.9 | 121.3 | 121.8 | 121.9 | 121.9 |
| 7 | 24.9 | 24.7 | 24.7 | 24.6 | 24.7 |
| 8 | 43.9 | 44.4 | 43.0 | 43.0 | 42.9 |
| 9 | 49.9 | 49.9 | 50.3 | 50.3 | 50.4 |
| 10 | 35.0 | 35.0 | 36.0 | 36.0 | 36.5 |
| 11 | 214.0 | 215.1 | 215.6 | 215.7 | 216.0 |
| 12 | 49.9 | 49.7 | 50.0 | 50.0 | 50.3 |
| 13 | 52.0 | 52.0 | 51.6 | 51.5 | 51.5 |
| 14 | 49.7 | 49.9 | 49.7 | 49.7 | 50.2 |
| 15 | 44.5 | 43.9 | 46.6 | 44.7 | 44.6 |
| 16 | 73.0 | 73.0 | 75.5 | 75.4 | 74.4 |
| 17 | 56.3 | 56.0 | 56.0 | 56.2 | 56.3 |
| 18 | 20.3 | 20.7 | 20.4 | 20.4 | 20.7 |
| 19 | 20.7 | 20.3 | 20.7 | 20.6 | 20.9 |
| 20 | 79.8 | 79.6 | 79.6 | 79.8 | 79.8 |
| 21 | 24.8 | 24.9 | 24.8 | 24.8 | 24.8 |
| 22 | 204.6 | 204.3 | 204.3 | 204.5 | 204.5 |
| 23 | 121.3 | 120.6 | 120.7 | 121.8 | 122.4 |
| 24 | 153.0 | 156.6 | 156.6 | 153.0 | 153.0 |
| 25 | 81.0 | 71.7 | 71.7 | 81.0 | 81.0 |
| 26 | 26.9 | 29.9 | 29.5 | 27.1 | 27.1 |
| 27 | 27.1 | 29.5 | 29.5 | 26.9 | 26.9 |
| 28 | 30.0 | 29.5 | 28.5 | 28.5 | 28.4 |
| 29 | 21.3 | 22.0 | 21.2 | 20.9 | 20.7 |
| 30 | 19.4 | 19.5 | 18.9 | 18.8 | 18.6 |
| 1' | | | | | 101.3 |
| 2' | | | | | 75.3 |
| 3' | | | | | 77.7 |
| 4' | | | | | 70.7 |
| 5' | | | | | 78.3 |
| 6' | | | | | 62.0 |
| Ac | 172.6 | 172.3 | 172.3 | 172.5 | 172.5 |
| Ac-Me | 21.9 | 21.2 | 20.9 | 21.3 | 21.3 |
| Ac | 171.8 | | | 171.8 | 171.8 |
| Ac-Me | 22.0 | | | 21.9 | 21.9 |

Compound 1 represent 16-O-acetyl CuB. Compound 2 represent 16-O-acetyl CuD. Compound 3 represent 16-O-acetyl CuI. Compound 4 represent 16-O-acetyl CuE. Compound 5 represent 16-O-acetyl CuE-Glu.

**Table 3 Enzyme kinetic parameters of ACT3.**

| Substrates | $K_m$ (µM) | $k_{cat}$ (S$^{-1}$) | $K_{cat}/K_m$ (M$^{-1}$S$^{-1}$) | $V_{max}$ (nmol/min/mg) |
|---|---|---|---|---|
| CuB | 11.75 ± 1.74 | 1.24 × 10$^{-2}$ | 10.56 × 10$^2$ | 10.34 ± 1.12 |
| CuD | 10.39 ± 1.19 | 1.83 × 10$^{-2}$ | 17.63 × 10$^2$ | 15.27 ± 1.17 |
| CuE | 7.63 ± 4.83 | 0.55 × 10$^{-2}$ | 7.21 × 10$^2$ | 4.58 ± 1.86 |
| CuE-Glu | 11.63 ± 1.62 | 2.47 × 10$^{-2}$ | 21.21 × 10$^2$ | 20.56 ± 2.06 |
| CuI | 8.62 ± 0.95 | 1.29 × 10$^{-2}$ | 14.97 × 10$^2$ | 10.75 ± 0.71 |

ACT3 enzyme activity was determined using variable concentrations of substrate at a fixed enzyme concentration at 30 °C. $V_{max}$ and $K_m$ values were obtained by fitting values to Michaelis–Menten kinetics. Values are means ± SD, $n = 3$.

cucurbitacin forms as sugar acceptors. Similar results have been reported in previous studies, in which UGT7AM3, a cucumber UGT, could only glycosylate CuC among other cucurbitacins sharing structural similarity[17].

**Accumulation of biosynthesized cucurbitacins.** Plant secondary metabolites, such as cucurbitacins, rapidly accumulate in damaged and adjacent leaves of *Cucurbitaceae* plants for protection against phytopathogens and insect herbivory[27–30]. Time courses of cucurbitacin accumulation in response to mechanical injury were compared to determine whether cucurbitacins newly acetylated by ACTs can be induced by wounding. HPLC analyses were performed to evaluate the levels cucurbitacins in wounded watermelon leaves. Interestingly, high accumulation

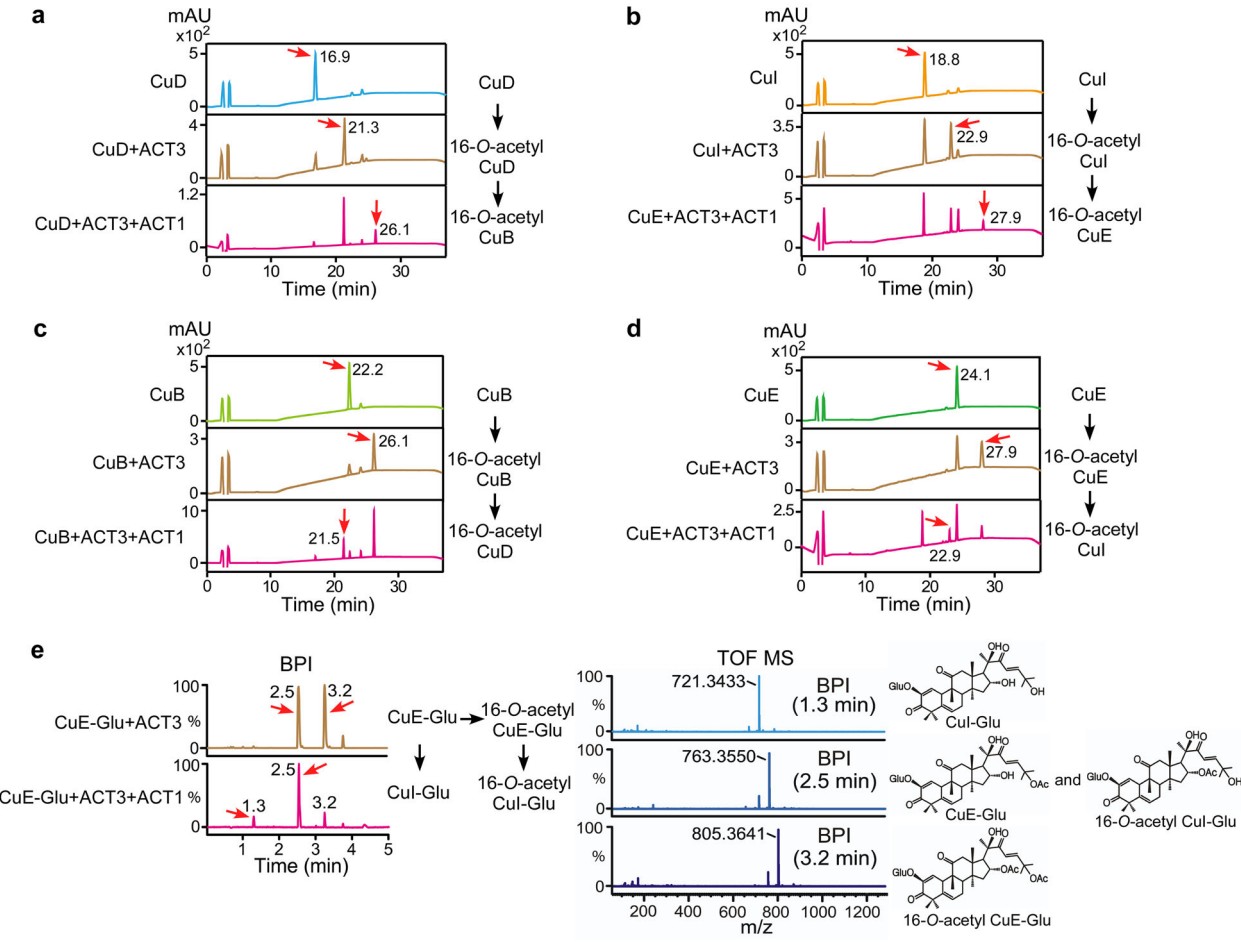

**Fig. 2 ACT1 catalytic activity in the biosynthesis of new compounds. a** HPLC analysis of sequential ACT3 and ACT1 enzymatic reactions in vitro using CuD as a primary substrate. The chromatogram shows a product peak at 21.3 min (for 16-*O*-acetyl CuD) in the primary reaction and at 26.1 min (for 16-*O*-acetyl CuB) in the secondary reaction. **b** HPLC analysis of sequential ACT3 and ACT1 enzymatic reactions in vitro using CuI as a primary substrate. The chromatogram shows a product peak at 22.9 min (for 16-*O*-acetyl CuI) in the primary reaction and at 27.9 min (for 16-*O*-acetyl CuE) in the secondary reaction. **c** HPLC analysis of sequential ACT3 and ACT1 enzymatic reactions in vitro using CuB as a primary substrate. The chromatogram shows a product peak at 26.1 min (for 16-*O*-acetyl CuB) in the primary reaction and at 21.5 min (for 16-*O*-acetyl CuD) in the secondary reaction. **d** HPLC analysis of sequential ACT3 and ACT1 enzymatic reactions in vitro using CuE as a primary substrate. The chromatogram shows a product peak at 27.9 min (for 16-*O*-acetyl CuE) in the primary reaction and at 22.9 min (for 16-*O*-acetyl CuI) in the secondary reaction. **e** LC-MS analysis of the extract prepared from sequential ACT3 and ACT1 enzymatic reactions in vitro using CuE-Glu as a primary substrate. The BPI chromatogram of the reaction product in negative-ion mode shows peaks at 2.5 and 3.2 min in the primary reaction and at 1.3, 2.5, and 3.2 min in the secondary reaction. Peaks in the extracted ion chromatogram at *m/z* 721.3433 [M+FA-H]⁻, 763.3550 [M+FA-H]⁻, and 805.3641 [M+FA-H]⁻ correspond to CuI-Glu, CuE-Glu/16-*O*-acetyl CuI-Glu, and 16-*O*-acetyl CuE-Glu, respectively. The structures of the product are presented on the right. Red arrows indicate reaction substrates and products.

of 16-*O*-acetyl CuE was observed 2 h after wounding treatment (Fig. 3b); however, the accumulation of other produces was undetectable or unchanged in wounded leaves. Once accumulated, 16-*O*-acetyl CuE was sustained at least for 18 h (Supplementary Fig. 16a, b). *ACT* gene expression was also evaluated by RT-qPCR analysis to determine the relationship with 16-*O*-acetyl CuE accumulation. *ACT* gene expression was also increased in similar a pattern to 16-*O*-acetyl CuE accumulation (Fig. 3c). To reveal the correlation between wound-induced cucurbitacins and the levels of ACT1, ACT2, and ACT3 translation in vivo, GFP-tagged *ACT1*, *ACT2*, and *ACT3* genes and control *GFP* were transiently expressed in watermelon melon, and the cucurbitacin contents were measured by HPLC analysis (Fig. 3d and Supplementary Fig. 16c–e). Accumulation of 16-*O*-acetyl CuE was only detectable in leaves transiently overexpressing *ACT3*. These results indicate that

*ACT3* is responsible for the in vivo accumulation of 16-*O*-acetyl CuE in response to wounding.

Wounding is preceded by herbivore invasion, which induces the accumulation of cucurbitacin and leads to changes in cucurbitacin profiles[30]. In addition, changes in ring A and the side chain of cucurbitacins have been shown to modify the antifeedant activity of herbivores[31]. Therefore, we investigated how changes in the cucurbitacin profile proportional to the accumulation of 16-*O*-acetylated derivatives following wounding, affected insect taste, especially bitter tastants of fruit flies. The main taste organ of *Drosophila* consists of bilaterally symmetrical labial palps, which are each decorated with stratified hair-like projections called sensilla. Each sensilla possesses a small pore at the tip, which enables neural activity to be analyzed[32].

To evaluate neuronal activation by CuB, 16-*O*-acetyl CuB, CuE, and 16-*O*-acetyl CuE, tip recording examinations were

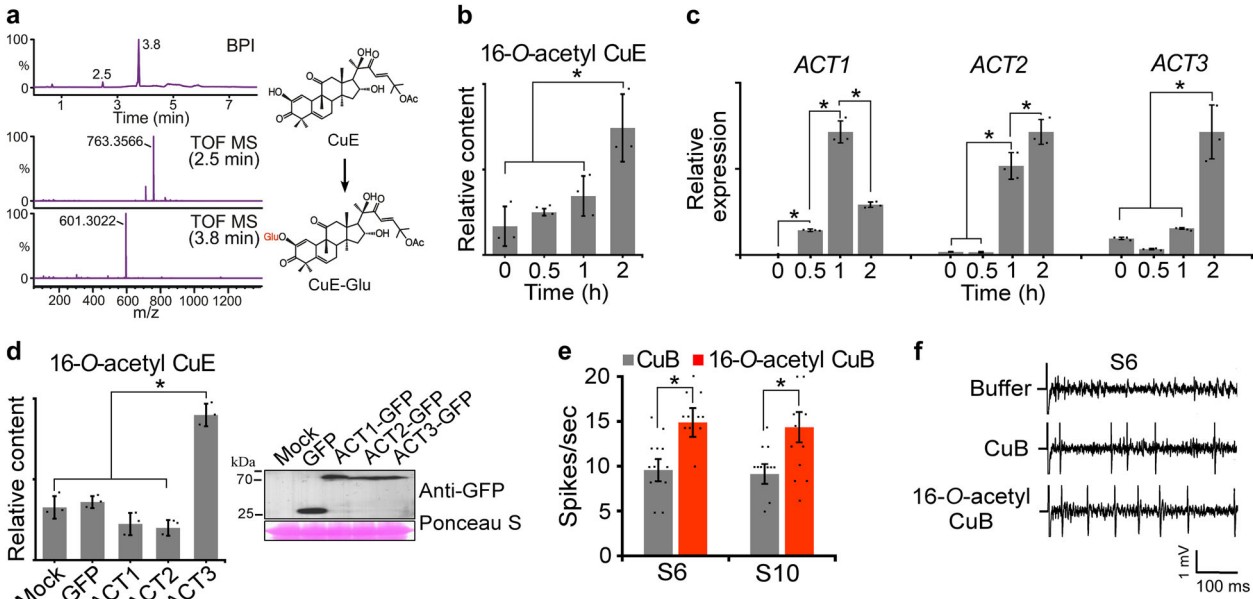

**Fig. 3 UGT74F2 catalytic activity and cucurbitacin accumulation in response to injury and transient *ACT3* overexpression.** Evaluation of neuronal activation against *Drosophila*. **a** LC-MS analysis of extracts prepared from enzymatic reactions of UGT74F2 in vitro using CuE as a substrate. BPI chromatogram of reaction product in negative-ion mode shows two peaks at 2.5 and 3.8 min. Peaks in the extracted ion chromatogram at *m/z* 763.3566 [M+FA-H]⁻ and 601.3022 [M+FA-H]⁻ correspond to CuE-Glu and CuE, respectively. The structures of the products (right) were elucidated using LC-MS and NMR spectroscopy. **b** Accumulation of 16-*O*-acetyl CuE in wounded leaves was determined by HPLC. **c** Results of RT-qPCR of *ACTs* from wounded leaves. **d** Immunoblot detection and accumulation of 16-*O*-acetyl CuE in leaves transiently overexpressing *ACT* genes. **e** Neuronal activation with CuB and 16-*O*-acetyl CuB. Average frequencies of action potentials elicited from S6 and S10 sensilla. ($n = 18$–$22$). **f** Representative sample traces obtained from S6 sensilla. Error bars represent ± SD ($n = 3$). Asterisks indicate significant differences (*$P < 0.05$).

performed using the S6 and S10 sensilla. These sensilla house bitter-sensing gustatory receptor neurons (GRNs), in which bitter-sensing gustatory receptors have aversive roles in feeding[32]. Although no differences in bitter recognition were observed by CuE acetylation, 16-*O*-acetyl CuB induced significantly higher levels of neuronal firing from S6 and S10 sensilla following application of 0.01 mM compound than CuB (Fig. 3e, f and Supplementary Figs. 16f, 17a–c). However, this difference was not observed at other tested concentrations. Furthermore, CuB was recently shown to directly activate bitter-sensing GRNs via GR33a[33]. This suggests that acetylation of cucurbitacins may alter taste recognition in herbivorous insects, leading to the attraction or repellence of these insects. Cucurbitacins act as chemical signals for specialized herbivores, such as behavioral arrestants and feeding stimulants[34–37]. Although no specific relationships were observed between herbivore species and specific types of cucurbitacins, modification of cucurbitacin structures induced a different response in herbivores[34,37]. Our findings showing the formation of divergent cucurbitacin derivatives by ACTs and UGT, the accumulation of specific types of cucurbitacin derivatives and *ACT* translation in response to wounding, provide information to address these questions.

## Discussion
Structurally, cucurbitacins are characterized by the tetracyclic cucurbitane backbone [19-(10 → 9β)-abeo-10α-lanost-5-ene], and differ from other tetracyclic triterpenes by the presence of numerous keto-, hydroxyl-, and acetoxy-groups (Fig. 4a)[9]. Ecdysone and cortisol were used for additional ACT promiscuity tests, because both contain the same structural triterpenoid backbone as cucurbitacins and are representative of insect and animal hormones. LC-MS analysis was conducted following three

ACTs in vitro enzymatic assays using ecdysone and cortisol as substrates. Surprisingly, all three ACTs catalyzed one acetylation on ecdysone (Fig. 4b) and cortisol (Fig. 4c), which shared substantial structural similarity to cucurbitacins. Considering that ACT2 demonstrated a lack of acetyltransferase activity against divergent cucurbitacin derivatives, it was surprising that ACT2 presented acetyltransferase activity toward both the substrates ecdysone and cortisol, which are not present in watermelon. This indicates that minor structural differences in substrates could affect the acetylation ability of these ACTs.

In this study, *ACTs* and *UGT* were identified from a comparative transcriptome analysis of watermelons. HPLC, LC-MS, and NMR spectroscopy were performed to investigate the functional role(s) of ACTs and UGT in the biosynthesis of cucurbitacin derivatives of watermelon. The results of an in vitro enzymatic assay revealed that ACT3 specifically acetylated C16 of CuB, CuD, CuI, CuE, CuI-Glu, and CuE-Glu (Fig. 1a–e and 4d, e). Unlike ACT3, ACT1 showed bifunctional activity as an acetyltransferase of CuD, CuI, 16-*O*-acetyl CuD, and 16-*O*-acetyl CuI and a deacetylase of CuB, CuE, 16-*O*-acetyl CuB, 16-*O*-acetyl CuE, CuE-Glu, and 16-*O*-acetyl CuE-Glu (Fig. 4d, e).

UGT74F2 preferred CuE as a sugar acceptor over the other cucurbitacin, producing CuE-Glu (Fig. 4e). A limitation of this study was that we were unable to identify the mechanism conferring ACT1, ACT2, and UGT74F2 with bifunctional or substrate specificity. In future, it will be interesting to investigate the bifunctional activity of ACT1, which acts as both an acetyltransferase and a deacetylase, and the substrate preference of ACT2 and UGT74F2 in the production of cucurbitacin derivatives through protein x-ray crystallography and enzymatic assay of partially mutated multiproteins. Although the results of an in vivo assay indicated that accumulation of 16-*O*-acetyl CuE was associated with the translation of *ACT3* in response to

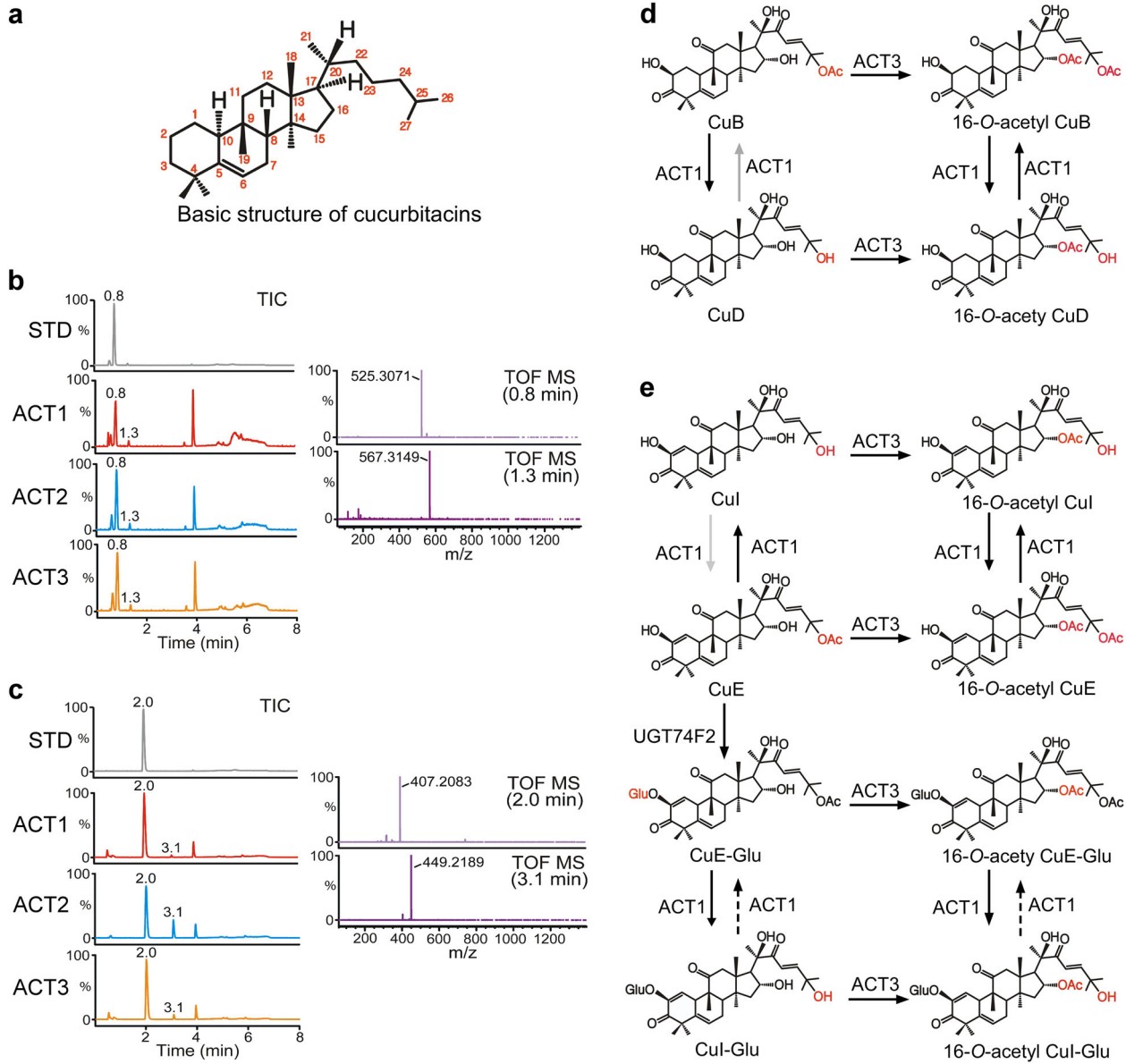

**Fig. 4 Additional catalytic activity. a** Basic structure of cucurbitacins. **b** LC-MS analysis of extracts prepared from enzymatic reactions of ACT1, ACT2, and ACT3 in vitro using ecdysone as a substrate. The total ion chromatogram (TIC) of the ecdysone standard (STD) and reaction product in negative-ion mode shows two peaks at 0.8 and 1.3 min. Peaks in the extracted ion chromatogram at $m/z$ 525.3071 [M+FA-H]$^-$ and 567.3149 [M+FA-H]$^-$ corresponds to ecdysone (calculated for $C_{28}H_{45}O_9$ 525.30639) and the ecdysone with the adducted acetyl group (calculated for $C_{30}H_{47}O_{10}$ 567.3169), respectively. **c** LC-MS analysis of cortisol STD and extract prepared from enzymatic reactions of ACT1, ACT2, and ACT3 in vitro using cortisol as a substrate. The TIC of the reaction product in negative-ion mode shows two peaks at 2.0 and 3.1 min. Peaks in the extracted ion chromatogram at $m/z$ 407.2083 [M+FA-H]$^-$ and 449.2189 [M+FA-H]$^-$ correspond to cortisol (calculated for $C_{22}H_{30}O_7$ 407.2070) and the cortisol with the additional acetyl group (calculated for $C_{24}H_{33}O_8$ 449.2176), respectively. **d** ACT1 acetylates the C25 hydroxyl moiety of CuD and 16-O-acetyl CuD and deacetylates the C25 acetyl moiety of CuB and 16-O-acetyl CuB, producing equivalent cucurbitacins. **e** ACT1 acetylates the C25 moiety of CuI and 16-O-acetyl CuI and deacetylates the C25 acetyl moiety of CuE and 16-O-acetyl CuE, producing equivalent cucurbitacins. ACT1 acetylates the C25 hydroxyl moiety of CuE-Glu and 16-O-acetyl CuE-Glu, resulting in CuI-Glu and 16-O-acetyl CuI-Glu. ACT3 acetylates the C16 hydroxyl moiety of CuI, CuE, CuE-Glu, and CuI-Glu, resulting in 16-O-acetyl CuI, 16-O-acetyl CuE, 16-O-acetyl CuE-Glu, and 16-O-acetyl CuI-Glu. UGT74F2 glucosylates CuE producing CuE-Glu. Line arrows indicate the conversions identified in this study, and dashed arrows indicate the unidentified conversions. Gray arrows indicate a previously revealed conversion.

wounding stress and transient overexpression, accumulation of other acetylated cucurbitacins (16-O-acetyl CuB, 16-O-acetyl CuD, and 16-O-acetyl CuI) induced by ACT3 was not detectable. This was because the induction of CuB, CuD, and CuI contents in watermelon leaves were too low to be detected in our analysis system. Nevertheless, these findings help to explain the responses of *Cucurbitaceae* plants to abiotic stress. ACT activity is not limited to the modification of cucurbitacin side chains, and may

extend to insect- and animal-originated triterpenoids with a similar core backbone.

## Methods

**Plant material.** Number One, *C. lanatus* germplasm, was used for cucurbitacin analysis, wounding, and transient expression. Seeds were kindly provided by Asia Seed Corporation, Seoul, Korea. Watermelon plants were grown in a growth chamber at 28 °C under long-day conditions (photoperiod, 16 h: 8 h, light: dark) at

a light intensity of 120 µmol m$^{-2}$ s$^{-1}$. Watermelon true leaves, 30–35 days after germination, were used for cucurbitacins analysis and wounding treatment. Leaves were wounded by crushing perpendicular to the mid-vein of the leaflet at intervals of 0.5 cm using a hemostat. The wounded and unwounded control plants were kept in separated boxes and incubated in a growth chamber.

**RNA expression analysis**. Gene sequences were collected from the watermelon genome database (http://cucurbitgenomics.org). Leaves were collected to examine the expression of *ACTs*. For RT-qPCR analysis, total RNA was extracted using a RiboEx Total RNA Kit (GeneAll). RNA quality was determined using a Nanodrop ND-2000 spectrophotometer (Nanodrop Technologies) and only high-quality RNA samples (A260/A230 > 2.0 and A260/A280 > 1.8) were used for subsequent experiments. RNA (5 µg) was used for complementary DNA (cDNA) synthesis, according to the manufacturer's instructions (Toyobo). RT-qPCR analysis was performed in 96-well plates with a CFX real-time system (Bio-Rad) using THUNDERBIRD SYBR pPCR mix (Toyobo). One stably expressed *TIP41* (*Cla016074*) gene was used as a reference, following "The eleven golden rules for quantitative RT-PCR"[38,39]. All RT-qPCR experiments were performed in two biological replicates (independently harvested samples) with three technical triplicates each. Oligonucleotide sequences used for the expression analysis are provided in Supplementary Table 3. To determine the relative abundance of transcripts, the data were analyzed using the Bio-Rad CFX Manager software (Bio-Rad).

**Cloning of *ACTs* and *UGT74F2* genes, and recombinant protein expression**. cDNA synthesis was performed with 5 µg of total RNA (prepared as described for the RNA expression analysis) extracted from watermelon vegetative tissues (leaves, stems, and roots) using SuperiorScript III Master Mix (Enzynomics) in accordance with the manufacturer's instructions. cDNA fragments (1 µL) were used as a PCR template with 36 bp forward and reverse primers (Supplementary Table 3) designed according to the manufacturer's instructions of EZ-Fusion™ cloning kit (Enzynomics). PCR was performed using i-Pfu™ polymerase (iNtRON). Amplified *ACTs* and *UGT74F2* DNA fragments were purified with MEGA quick-spin™ plus a total fragment DNA purification kit (INtRON), and then cloned into the pET-28a vector according to the manufacturer's instructions of the EZ-Fusion™ cloning kit (Enzynomics). The resulting vectors were then transformed into *E. coli* cells (BL21 DE3) for protein expression. A single colony was selected using colony PCR and then cultured in 5 mL of medium overnight. After 16 h, seed cells were inoculated in 1 L of selective liquid medium at 37 °C and grown until the OD$_{600}$ value was within the range 0.4–0.6. After 1 mL of 1 M IPTG was added to medium for protein expression, cells were grown for a further 16 °C at 16 h, after which they were collected by centrifugation (4000 × $g$, 15 min, 4 °C).

**Purification of recombinant His-tagged recombinant proteins**. Cell pellets were re-suspended in 30 mL binding buffer (0.5 M NaCl and 20 mM sodium phosphate, pH 7.4), and then cells were fragmented by sonication (Sonics & Materials Inc.). Supernatants were recovered by centrifugation (25,000 × $g$, 15 min) and mixed with a Ni Sepharose™ 6 Fast Flow resin (GE healthcare) and stirred at 4 °C for 1 h in a column. After stirring, the resin was washed sequentially with three volumes of binding buffer and washing buffer (100 mM imidazole, 0.5 M NaCl, 20 mM sodium phosphate, pH 7.4). Then, the column was eluted with three volumes of elution buffer (200 mM imidazole, 0.5 M NaCl, 20 mM sodium phosphate, pH 7.4). Elution buffer containing purified proteins was exchanged with 0.1 M sodium phosphate buffer (pH 7.4), concentrated to a final volume 250 µL using 30-kDa Amicon® Ultra-15 centrifugal filters (Millipore), and stored as 10% glycerol stock at −80 °C. The protein concentration was measured using a Bradford protein assay reagent kit (Biosesang).

**In vitro enzymatic assay and assessment of enzymes kinetics**. CuB, CuD, CuE, CuI, and CuE-Glu were purchased from Phytolab. To confirm the activity of ACT proteins, 40 µg of purified protein and 4 µL of 10 mM acetyl Co-A were dissolved in 50 mM of sodium phosphate buffer (pH 7.4), and 2 µL of each 10 mg/mL cucurbitacin standard (CuB, CuD, CuE, CuE-Glu, and CuI) was added to the solution, and made up to 200 µL with 50 mM sodium phosphate buffer (pH 7.4). The reaction solutions were incubated at 30 °C for 1 h. Then, the solutions were mixed with 200 µL of ethyl acetate by vortexing, and the supernatant was recovered by centrifugation (1000 × $g$, 10 s). The ethyl acetate extraction step was repeated three times and supernatants were pooled. To confirm the activity of UGT proteins, 10 µg of purified protein and 2 µL of 100 mM UDP-glucose were dissolved in 50 mM sodium phosphate buffer (pH 7.4). Then, 0.5 µL of 2 M MgCl$_2$ and 2 µL each of 10 mg/mL cucurbitacin aglycone standards (CuB, CuD, CuE, and CuI) were placed into solution and made up to 200 µL with 50 mM sodium phosphate buffer (pH 7.4). Reactions and acetate extractions were performed as described above. Supernatants recovered from the ACT and UGT enzymatic assays were dried by a speed vacuum concentrator (Eyela), and then reconstituted with MeOH prior to injection into the HPLC and/or LC-MS and NMR spectroscopy. Kinetic studies were performed in 100 µL volumes with 20 µg of purified recombinant protein, 10 mM acetyl Co-A, 50 mM sodium phosphate (pH 7.4), and cucurbitacins (41.7–93.2 µM). Mixtures were incubated at 30 °C for 30 min and the reaction was stopped by adding 100 µL ethyl acetate; the ethyl acetate extraction step was repeated two more times. All subsequent procedures were performed as described

above. Kinetic parameters were determined from triplicate Michaelis–Menten kinetic experiments.

**Analytical conditions for HPLC analysis**. Collected watermelon samples were ground to a powder in a mortar with liquid nitrogen. The samples (0.3 g) were homogenized in 1 mL of 100% methanol (MeOH), sonicated for 1 h, and incubated at 55 °C for 1 h. The supernatants were recovered by centrifugation at 13,000 × $g$ for 10 min, and then filtered through a 0.45 µm polyvinylidene difluoride (PVDF) membrane (Pall). The extracts were concentrated in a speed vacuum concentrator (Eyela) for 1 day, and then reconstituted in 120 µL MeOH prior to injection into the HPLC system. The HPLC system was composed of a LC-20AD pump, a SPD-20A UV/VIS detector, a CBM-20A communications bus module, and a SIL-20AC autosampler (Shimadzu), which was used in combination with a 4.6 × 300 mm Syncronis C$_{18}$ column (Thermo Scientific). HPLC separation of tissue extracts and enzymatic assay products was performed using the same mobile phases, consisting of two solvents: (A) H$_2$O with 0.1% formic acid and (B) acetonitrile with 0.1% formic acid. The samples were injected at a total volume of 20 µL, and the flow rate of the solvent was maintained at 1 mL/min. The sample was separated using the following gradient elution profile: 30% B for 5 min, which was increased to, and maintained at, 70% B for 20–30 min, and then restored to, and maintained at, 30% B for 40–45 min. The UV signals were monitored at 230 nm. For NMR spectroscopy samples, additional HPLC fraction steps were added to enrich and obtain NMR spectroscopy-grade products. The composition and conditions of the HPLC system were as described above, except for the column (YMC-Trial C$_{18}$ ExRS, YMC), flow rate (4 mL/min), and injection volume (100 µL). The compounds fractionated from ACT and UGT enzymatic reactions were dried by a speed vacuum concentrator (Eyela) prior to NMR spectroscopy. NMR spectra were recorded on a Bruker Avance 600 (Billerica).

**LC-MS**. Chromatographic separation was performed using an ultra-performance liquid chromatography/quadrupole time-of-flight mass spectrometer (UPLCQ/TOF-MS, Synapt G2Si, Waters). UPLC separation was performed using an Acquity BEH C$_{18}$ column (2.1 mm × 100 mm, 1.7 µm, Waters). Mobile phase A consisted of H$_2$O containing 0.1% formic acid, whereas mobile phase B contained 0.1% formic acid in acetonitrile. Samples were eluted using the following conditions: initial 30% B to 70% at 4 min, to 70% B at 6 min, to 30% B at 6.3 min, and equilibrated for an additional 1.7 min. The flow rate was 0.4 mL/min. Column temperature was maintained at 40 °C. Mass acquisition was performed in negative electrospray ionization mode, with the following parameters: capillary voltage of 2.0 kV, cone voltage of 10 V, source temperature of 110 °C, desolvation temperature of 450 °C, and desolvation gas flow of 900 L/h. Mass data were collected with a scan time of 0.25 s.

**Transient expression and immunoblotting of ACTs**. The coding regions of *ACTs* were amplified and cloned into the pCAMBIA1300 binary vector downstream of the *35S* promoter. Green fluorescent protein (GFP) was linked to the promoter as a reporter gene. cDNA fragments (1 µL) were used as a PCR template with 36 bp forward and reverse primers (Supplementary Table 3) designed according to the instructions of the EZ-Fusion™ cloning kit (Enzynomics). After sequencing the conformation of *ACTs* fused to the binary vector, the plasmids were introduced into *Agrobacterium tumefaciens* strain GV3101 using the freezing transformation method. Infiltration was performed as described earlier[40] and agro cells were infiltrated into the first and second true leaves of 6-week-old watermelon plants. Leaf samples were collected after 2-days infiltration for use in the expression assay and for HPLC analysis. GFP-tagged ACT proteins were then extracted and detected as previously described[41]. Extracted proteins were separated by SDS-PAGE and transferred to Amersham™ Hybond™ polyvinylidene fluoride (PVDF) membrane (GE Healthcare). GFP-tagged ACTs were detected using rabbit polyclonal antibodies against GFP (Santa Cruz) and anti-rabbit IgG horseradish peroxidase (HRP) (Thermo Scientific). HRP was detected using Super Signal West Pico (Pierce) and X-ray film (Agfa).

***Drosophila* stock and electrophysiology**. Flies were obtained from the *Drosophila* stock center (https://bdsc.indiana.edu/), and $w^{1118}$ flies were used as the wild-type control. Tip recording examinations were performed as previously described[42]. We immobilized 3–7-day-old flies by keeping them on ice and then inserted reference glass electrodes filled with Ringer's solution into the thorax of flies, extending the electrode toward their proboscis. We stimulated the sensilla with a tastant dissolved in 1 mM KCl solution in recording pipettes (10–20 µm tip diameter). The recording electrode was connected to a preamplifier (Taste PROBE, Syntech), and the signals were collected and amplified 10×, using a signal connection interface box (Syntech) with a 100–3000 Hz band-pass filter. Recordings of action potentials were acquired using a 12 kHz sampling rate and analyzed using Autospike 3.1 software (Syntech).

**Elucidation of compound structure by MS and NMR spectroscopy data analysis**. 16-*O*-acetyl cucurbitacin B (compound 1) was determined to be C$_{34}$H$_{48}$O$_9$ based on the negative ESI-MS $m/z$ 645 [M+formic acid-H]$^-$; negative HR ESI-MS $m/z$ 645.3282 [M+formic acid-H]$^-$ (calculated for C$_{35}$H$_{49}$O$_{11}$ 645.3275); $^1$H-NMR (600 MHz, CD$_3$OD, δ$_H$) and $^{13}$C-NMR (150 MHz, CD$_3$OD, δ$_C$) in Tables 1 and 2. 16-*O*-acetyl cucurbitacin D (compound 2) was determined to be C$_{32}$H$_{46}$O$_8$ based on

on the negative ESI-MS $m/z$ 603 [M+formic acid-H]$^-$; negative HR ESI-MS $m/z$ 603.3195 [M+formic acid-H]$^-$ (calculated for $C_{33}H_{47}O_{10}$ 603.3169); $^1$H-NMR (600 MHz, CD$_3$OD, $\delta_H$) and $^{13}$C-NMR (150 MHz, CD$_3$OD, $\delta_C$) in Tables 1 and 2. 16-$O$-acetyl cucurbitacin I (compound 3) was determined to be $C_{32}H_{44}O_8$ based on the negative ESI-MS $m/z$ 601 [M+formic acid-H]$^-$; negative HR ESI-MS $m/z$ 601.3026 [M+formic acid-H]$^-$ (calculated for $C_{33}H_{45}O_{10}$ 601.3012); $^1$H-NMR (600 MHz, CD$_3$OD, $\delta_H$) and $^{13}$C-NMR (150 MHz, CD$_3$OD, $\delta_C$) in Tables 1 and 2. 16-$O$-acetyl cucurbitacin E (compound 4) was determined to be $C_{34}H_{46}O_9$ based on the negative ESI-MS $m/z$ 643 [M+formic acid-H]$^-$; negative HR ESI-MS $m/z$ 643.3126 [M+formic acid-H]$^-$ (calculated for $C_{35}H_{47}O_{11}$ 643.3118); $^1$H-NMR (600 MHz, CD$_3$OD, $\delta_H$) and $^{13}$C-NMR (150 MHz, CD$_3$OD, $\delta_C$) in Tables 1 and 2. 2-$O$-$\beta$-D-glucopyranosyl 16-$O$-acetyl cucurbitacin E (compound 5) 5 was determined to be $C_{40}H_{56}O_{14}$ based on the negative ESI-MS $m/z$ 759 [M-H]$^-$, 805 [M+formic acid-H]$^-$; negative HR ESI-MS $m/z$ 805.3693 [M+formic acid-H]$^-$ (calculated for $C_{41}H_{57}O_{16}$ 805.3646); $^1$H-NMR (600 MHz, CD$_3$OD, $\delta_H$) and $^{13}$C-NMR (150 MHz, CD$_3$OD, $\delta_C$) in Tables 1 and 2.

**Statistics and reproducibility**. All data were subjected to analysis of a $t$ test using R version 3.6.1 software. The data were represented as mean values ± standard error using three replicates. The differences between the samples were determined at $P < 0.05$ level.

**Reporting summary**. Further information on research design is available in the Nature Research Reporting Summary linked to this article.

## Data availability

Source data for all figure are included as Supplementary data. The data that support the findings of this study are available from corresponding author upon reasonable request.

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

## Acknowledgements

We thank Korea Basic Science Institute (KBSI) for providing UPLCQ/TOF-MS data. This work was supported under basic science research program (NRF 2017R1D1A1A02018460 and 2020R1A2C108800) funded by National Research Foundation of Korea and the New Breeding technologies development Program (Project No. PJ014872012020) of the Rural Development Administration, Republic of Korea.

## Author contributions

Y.-C.K. and S.L. designed the research. Y.-C.K., D.C., and A.C. performed the majority of research. Y.-G.L. and N.-I.B. analyzed NMR data. S.R., J.S., and Y. L. performed and analyzed *Drosophila* experiments. Y.-C.K, S.L., N.-I.B., and Y. L. commented on the results. Y.-C.K. and S.L. combined the results and wrote the paper.

## Competing interests

The authors declare no competing interests
