## [Peer Review File · Communications Biology]

Reviewers' comments:

Reviewer #1 (Remarks to the Author):

This is a very impressive contribution to cucurbitacin biosynthesis and the structure activity relationships with respect to insect herbivore perception.

It is not totally clear to me if the UGTs and the P450s have been officially named by the two respective nomenclature committees.

In the material and methods, most of the times the degree Celsius is missing and there are many cases of double spacing.

There are several places where the English needs a bit more attention. Especially in the abstract. eg please have a closer look at sentences at line 31-32 and 33-35. Also line 21 do the authors mean tastant and not tastepant? I guess an "a" or (s) is missing in the title: Elucidation of missing link in biosynthesis of cucurbitacin derivatives and their 2 biological importance.

all the best

Reviewer #2 (Remarks to the Author):

The study of Kim et al., 'Elucidation of missing link in biosynthesis of cucurbitacin derivatives and their biological importance', describes the identification of enzymes, acetyltransferases (ACTs) and a UDP-glucosyltransferase (UGT), involved in synthesis of various acetylated cucurbitacin derivatives in watermelon. The authors use RNA-sequencing of two watermelons to select candidate genes coding for enzymes that catalyze modifications of cucurbitacins. They subsequently characterize the enzymatic activities of four of the 16 candidate enzymes (3 different ACTs and one UGT) by HPLC, LC-MS, NMR, and an in vitro enzymatic assay. The authors conclude that (i) ACT3 acetylates specifically cucurbitacins at C16, (ii) ACT2 shows no activity at all, and (iii) ACT1 functions both as acetyltransferase to some cucurbitacin derivatives and as deacetylase to others. The authors further suggest that the studied UGT, UGT74F2, prefers one specific cucurbitacin derivative. Finally, they perform an in vivo assay employing *Drosophila* and conclude that acetylation of cucurbitacin increases neuronal activity in insects, indicating a reaction of herbivore species to cucurbitacin derivatives.

I appreciate the application of various analytical methods to characterize the cucurbitacin derivatives. The study addresses a specific question for experts of biosynthesis of bitter substances in (some specific) plants. I miss significance and relevance for scientists working in adjacent fields. Beyond cucurbitacin biosynthesis research in plants, I think there is only little relevance for other fields. Overall the manuscript does not make a good impression; in addition to numerous shortcomings regarding grammar and language throughout the manuscript, the text lacks a thorough introduction of the topic and detailed explanations of data interpretation in a number of cases. Furthermore, it is difficult, if possible at all, to evaluate the experimental basis in case of the NMR studies for which the data are presented in poor quality in the Supplementary Information. It is also difficult to assess the RNA-seq analysis (see specific point 10 below). I am not sure if these analyses and the in vivo experiment are technically sound.

I think the manuscript, and the presentation of the data and conclusions need to be improved significantly. Some suggestions that should help are the following points:

- 1) The manuscript needs thorough reworking regarding language and grammar throughout (and the SI as well). As this addresses an editorial issue, I refrain from detailing the (many) specific sentences that need to be edited.
- 2) I disagree that the authors show the biological importance of their findings conclusively; the title and respective statements in the abstract (lines 37–38) and conclusions (lines 197–202) should thus be changed.
- 3) It is confusing that some methods are in the SI, some in the main text.
- 4) Line 21: I think 'tastepant', whatever it means, is a very specific term and should be explained.
- 5) Line 22: I don't understand what 'using cytotoxicity' means in this context.
- 6) Line 24: What specifically do the authors mean by 'link(s)'?
- 7) Lines 29–33: The authors state that they observe specific enzymatic activities, but also conclude that ACTs modify other triterpenoids as well. This seems to be contradictory and should be explained in more detail.
- 8) Lines 37–38: I don't really understand this sentence, I think it needs to be re-phrased.
- 9) Lines 42–50: Here, only limited background information is provided to introduce the topic and specific questions addressed in the study. It would help to address questions like (as examples): What is the role and function of cucurbitacins in plants? Where do they occur: in all plants, cell types, organelles? What are similar substances and related functions in other organisms? Why is it important and interesting to study the biosynthesis pathway? What are relevant implications of their findings for other researchers? Are there other modifications than acetylation? There is also no introduction to the different derivatives (e.g. CuE, CuB, CUI, CuD...). It is difficult to understand the topic given the limited information provided here. This is basically the only introductory paragraph, and I had a hard time to get enough background as a non-cucurbitacin-plant researcher. In particular, the significance of the study doesn't become clear.
- 10) Lines 51–59: Based on how the RNA-seq analysis is presented, it is not clear if there is any experimental evidence for selecting specifically these genes. What are 'WM3 and WM30' and why are these suitable for observing different cucurbitacin profiles? Are these different individuals or two different germplines (e.g. knock-out)? Only citing a previous study is not sufficient, the reader cannot understand the fundamentals of the experimental design. It should be explained what the total number of reads means. Why is CYP450 mentioned here (it is not studied anyway)? What is the basis for selecting these 16 genes? Why are these four specific ones then selected for further analyses and no other ones that show high log₂FC in SI Table 2 (e.g. CPQ, CYP87A3, crtN, UGT90A1 (and other UFTs that show even higher log₂FC than the selected UGT74F2)? The legend of SI Table 2 also needs more information and CYP87A3 is listed twice without any explanation. Why is the plant material for the RNA-seq experiments of different origin than the one for the qRT-PCR studies (Asia Seed Corp. versus USDA GRIN)? Is this then comparable at all (e.g. is the genomic profile in both cases the same)? This needs to be addressed by broader sequencing analyses. It would be good to provide fold changes also in SI Fig 1 for better comparison. Taken together, these fundamental analyses are not convincing in the current state, and it should be presented more conclusively.
- 11) Line 60: Again, why these four genes? Others show even higher fold changes in the RNA-seq data.

It also is not quite clear why soluble properties are adequate as a criterion at this point (should be explained better).

12) Line 67: What is UGT73AM3?

13) Lines 79–80: All NMR data should be presented such that each spectrum can be evaluated by the reader. For example, in all NMR-related Figures, correct axes labels are missing and numbers are very difficult to read, signals cannot be identified as the spectra are plotted too small, peaks are not labeled with the respective atom, the basis for chemical shift assignments is not clear (are there previous studies?), and color-codes are not explained. Especially the assignment procedure has to be presented such that everyone can follow it.

14) Line 84: How do the authors know that C16 is acetylated by ACT3? Has this been shown in previous studies (no citation is given here)? If not, I strongly suggest showing this by NMR.

15) Line 90: What is the evidence for the sugar moiety playing a major role in catalytic efficiency?

16) Line 91: From which source do these leaves originate? This is not clear from the Methods section.

17) Line 102: Based on the background information provided by the authors, I was very surprised that ACT2 does not show any enzymatic activity. What is the reason for this? Is this reproducible? The authors should at least include a discussion about it; I miss (possible) explanations for this observation.

18) Line 102–120: I was even more surprised that ACT1 acts both as acetylase and deacetylase. It does not become clear how this is possible. Is this observed for other acetyltransferases as well? I strongly suggest that further analyses are included towards establishing a mechanistic explanation of this finding. Or is there strong evidence in precedent literature? A possible starting point could be to investigate whether multiple active centers in ACT1 exist. As long as no mechanism is provided, this conclusion is not convincing.

19) Lines 121–129: An explanation why UGT74F2 glycosylates CuE, but not CuB, CuD and CuI is missing.

20) Line 136–138: I don't understand why only 16-O-acetyl CuE was detected in wounded leaves and not any of the other derivatives, especially as expression of all three ACTs seems to be increased (Fig 3c). This doesn't make sense to me.

21) Lines 149–158: I appreciate that the authors provide quite some background for the in vivo experiment. Some careful editing regarding language can improve this section further.

22) Lines 159–167: Why do the authors choose *Drosophila* as a model system (why is it suitable for this experiment)? Why does 16-O-CuE not enhance neuronal activity in *Drosophila*, even though it was found to be increased by ACT3 in wounded leaves (Fig 3a-d)? Also, the authors conclude that 16-O-acetyl CuB increases neuronal activity (Fig 3e,f and lines 159–161), but they stated that only 16-O-acetyl CuE is found in wounded leaves (lines 136–138). This suggests that the 16-O-CuE stimulation is not relevant. I further think the authors should perform the tip recordings at various 16-O-acetyl CuB concentrations to validate the significance of the observed neuronal stimulation (I find only data for 0.01 mM). What are physiologically relevant concentrations? In addition, the authors should show more examples of recordings as in Fig 3f, for example in the SI, and present them with a longer trajectory over time. The average frequency plot is based on numbers per sec, but only a shorter time

window, and only one trajectory per compound is shown in Fig 3f. What is the evidence that the observed increase in neuronal activity is really due to taste? Is it not conceivable, in particular given the small differences observed (Fig 3e), that other effects (e.g. conductivity of the solution) contribute to this increase? Is the receptor in *Drosophila* for these substances known? In summary, I feel that any biological relevance of the newly identified cucurbitacin derivatives is not evidenced by the experiments in the current state of the study.

23) Lines 171–186: I don't quite understand the point of performing these studies. I don't think there is strong experimental evidence in these analyses supporting the hypothesis that ACT acetylation is an evolutionarily conserved mechanism. Also, ACT2 did not show any activity for the tested plant cucurbitacins, but now for ecdysone and cortisol (Fig 4b,c). How can this be explained? I am skeptical that ACT2, being an enzyme from plants, catalyzes acetylation of triterpenoid of animals, but not of plants. Is the missing ACT2 enzymatic activity on the plant cucurbitacins really reproducible? Can the authors provide any further experimental evidence?

24) The legend of Fig 4d is missing.

I hope addressing these concerns help the authors to improve their manuscript and the presentation of some of their findings, to work towards a more conclusive and convincing study.

Referee expertise:

Referee #1: genomics, cucurbitacins

Referee #2: NMR, RNA-seq

Reviewers' comments:

Reviewer #1 (Remarks to the Author):

This is a very impressive contribution to cucurbitacin biosynthesis and the structure activity relationships with respect to insect herbivore perception.

Its is not totally clear to me if the UGTs and the P450s have been officially named by the two respective nomenclature committees.

Response: *Since some gene name are nomenclatural names and some of them are not, we added the gene ID of every UGTs and P450s showed in this study.*

In the material and methods, most of the times the degree Celsius is missing and there are many cases of double spacing.

Response: *We corrected Celsius degreed symbols.*

There are several places where the English needs a bit more attention. Especially in the abstract. eg please have a closer look at sentences at line 31–32 and 33–35. Also line 21 do the authors mean tastant and not tastepant? I guess an "a" or (s) is missing in the title: Elucidation of missing link in biosynthesis of cucurbitacin derivatives and their 2 biological importance.

Response: *We eliminated and changed the sentences at line 31–35 and 33–35 in abstract. We exchanged "tastepant" to "tastant" in text (line 203 and 400). We corrected the title "Elucidation of missing link in biosynthesis of cucurbitacin derivatives and their biological importance" to "Elucidation of missing links in biosynthesis of cucurbitacin derivatives and their biological importance" (line 1).*

all the best

Reviewer #2 (Remarks to the Author):

The study of Kim et al., 'Elucidation of missing link in biosynthesis of cucurbitacin derivatives and their biological importance', describes the identification of enzymes, acetyltransferases (ACTs) and a UDP-glucosyltransferase (UGT), involved in synthesis of various acetylated cucurbitacin derivatives in watermelon. The authors use RNA-sequencing of two watermelons to select candidate genes coding for enzymes that catalyze modifications of cucurbitacins. They subsequently characterize the enzymatic activities of four of the 16 candidate enzymes (3 different ACTs and one UGT) by HPLC, LC-MS, NMR, and an in vitro enzymatic assay. The authors conclude that (i) ACT3 acetylates specifically cucurbitacins at C16, (ii) ACT2 shows no activity at all, and (iii) ACT1 functions both as acetyltransferase to some cucurbitacin derivatives and as deacetylase to others. The authors further suggest that the studied UGT, UGT74F2,

prefers one specific cucurbitacin derivative. Finally, they perform an in vivo assay employing *Drosophila* and conclude that acetylation of cucurbitacin increases neuronal activity in insects, indicating a reaction of herbivore species to cucurbitacin derivatives.

I appreciate the application of various analytical methods to characterize the cucurbitacin derivatives. The study addresses a specific question for experts of biosynthesis of bitter substances in (some specific) plants. I miss significance and relevance for scientists working in adjacent fields. Beyond cucurbitacin biosynthesis research in plants, I think there is only little relevance for other fields. Overall the manuscript does not make a good impression; in addition to numerous shortcomings regarding grammar and language throughout the manuscript, the text lacks a thorough introduction of the topic and detailed explanations of data interpretation in a number of cases. Furthermore, it is difficult, if possible at all, to evaluate the experimental basis in case of the NMR studies for which the data are presented in poor quality in the Supplementary Information. It is also difficult to assess the RNA-seq analysis (see specific point 10 below). I am not sure if these analyses and the in vivo experiment are technically sound.

I think the manuscript, and the presentation of the data and conclusions need to be improved significantly. Some suggestions that should help are the following points:

1) The manuscript needs thorough reworking regarding language and grammar throughout (and the SI as well). As this addresses an editorial issue, I refrain from detailing the (many) specific sentences that need to be edited.

Response: *Thank you for pointing your comments. Our manuscript was proofread by professional proofreading company. We also attached certification of editing in the last page.*

2) I disagree that the authors show the biological importance of their findings conclusively; the title and respective statements in the abstract (lines 37–38) and conclusions (lines 197–202) should thus be changed.

Response: *We eliminate the statements of line 37–38 in the abstract and changed the content of lines 197–202 (line 253–256)*

3) It is confusing that some methods are in the SI, some in the main text.

Response: *Methods for main results showed in figures (plant material for wounding, ACTs transient expression, RNA expression level. Methods related ACTs and UGT enzyme reaction, HPLC and LC-MS conditions, ACTs transient expression, and *Drosophila* electrophysiology) are in main text. Methods for SI results (plant material for RNA-seq, RNA-sequencing methods, NMR analysis, and semi RT-PCR) are in SI text.*

4) Line 21: I think 'tastepant', whatever it means, is a very specific term and should be explained.

Response: *We exchanged "tastepant" to "tastant" (line 199 and 396).*

5) Line 22: I don't understand what 'using cytotoxicity' means in this context.

Response: *We eliminated the statement of line 22.*

6) Line 24: What specifically do the authors mean by 'link(s)'?

Response: *We eliminated the statement of line 24.*

7) Lines 29–33: The authors state that they observe specific enzymatic activities, but also conclude that ACTs modify other triterpenoids as well. This seems to be contradictory and should be explained in more detail.

Response: *We eliminated the statement of line 29–33.*

8) Lines 37–38: I don't really understand this sentence, I think it needs to be re-phrased.

Response: *We eliminated the statement of line 37–38.*

9) Lines 42–50: Here, only limited background information is provided to introduce the topic and specific questions addressed in the study. It would help to address questions like (as examples): What is the role and function of cucurbitacins in plants? Where do they occur: in all plants, cell types, organelles? What are similar substances and related functions in other organisms? Why is it important and interesting to study the biosynthesis pathway? What are relevant implications of their findings for other researchers? Are there other modifications than acetylation? There is also no introduction to the different derivatives (e.g. CuE, CuB, CUI, CuD...). It is difficult to understand the topic given the limited information provided here. This is basically the only introductory paragraph, and I had a hard time to get enough background as a non-cucurbitacin-plant researcher. In particular, the significance of the study doesn't become clear.

Response: *We agree with this comment. Actually, this manuscript reviewed was submitted to "Nature plants letter" first, and then transferred to "communications biology" automatically on recommendation of Nature plants' editor. We could not provide background information about cucurbitacins, because there is limitation of words number and format (introduction part was not included). We added more information for cucurbitacins in the introduction (line 30–69).*

10) Lines 51–59: Based on how the RNA-seq analysis is presented, it is not clear if there is any experimental evidence for selecting specifically these genes. What are 'WM3 and WM30' and why are these suitable for observing different cucurbitacin profiles? Are these different individuals or two different germplines (e.g. knock-out)? Only citing a previous study is not sufficient, the reader cannot understand the fundamentals of the experimental design. It should be explained what the total number of reads means. Why is CYP450 mentioned here (it is not studied anyway)? What is the basis for selecting these 16 genes? Why are these four specific ones then selected for further analyses and no other ones that show high log₂FC in SI Table 2 (e.g. CPQ, CYP87A3, crtN, UGT90A1 (and other UFTs that show even higher log₂FC than the selected UGT74F2)? The legend of SI Table 2 also needs more information and CYP87A3 is listed twice without any explanation. Why is the plant material for the RNA-seq experiments of different origin than the one for the qRT-PCR studies (Asia Seed Corp. versus USDA GRIN)? Is this then comparable at all (e.g. is the genomic profile in both cases the same)? This needs to be addressed by broader sequencing analyses. It would be good to provide fold changes also in SI Fig 1 for better comparison. Taken together, these fundamental analyses are not convincing in the current state, and it should be presented more conclusively.

11) Line 60: Again, why these four genes? Others show even higher fold changes in the RNA-seq data. It also is not quite clear why soluble properties are adequate as a criterion at this point (should be explained better).

Response for 10) and 11): Transcriptome analysis was actually separated experiment for screening of cucurbitacin biosynthesis related genes. There are two kinds of representative watermelon germplasm (*Citrullus* spp.) including *Citrullus colocynthis* and *Citrullus lanatus*. *Citrullus colocynthis* is the wild ancestor of the modern watermelon (*Citrullus lanatus*). First, we selected two different individual watermelon (*Citrullus lanatus*), PI5322627 (USDA ID) containing the relatively high concentration of cucurbitacin and PI536451 (USDA ID) containing relatively low concentration of cucurbitacin (kim et al, Hort. Environ. Biotechnol. 2018 59:557–566) among diverse watermelon for transcriptome analysis. Because the number of the two watermelon (PI5322627 and PI536451) seeds offered from USDA were limited (about 10~15) and it will take some time for getting the enough amplified seed for additional works, we used “Number One” germplasm (*Citrullus lanatus*) which was commercially available and same species with PI5322627 and PI536451 for additional study. We changed “WM3 and WM30” to “PI5322627 (USDA ID) and PI536451 (USDA ID) in main text (line 83–84 and SI line 11–12)”, and added reason why we selected these two watermelons (line 83–87). We added explanations about the terms in SI table 1 (SI line 291–292).

For selecting the gene in SI table 2, we sorted the genes that up-regulated more than 1 of log₂FC in PI5322627 compare to PI536451 and described as cytochrome P450, UGT and acetyltransferase. After reconfirming the up-regulated genes in PI5322627 sample by RT-qPCR, we listed the genes in SI table 2. We added more information and explanation of the reason why BAHD1 and CYP87A3 are listed twice in SI table 2 (SI line 294–295 and 296–297). We believe that it is important to know which CYP450s are related in cucurbitacin biosynthesis pathway, and we tried. However, revealing the role of CYP450s in SI table 2 required more complicated experimental approach or methods, because the biosynthesis pathway for diverse cucurbitacins are little known until now, and intermediate products on the synthesis process also have been unknown. To find the role of CYP450s involved in unknown step of cucurbitacin biosynthesis pathway, yeast engineering that make yeast harboring multi genes are required, and additionally mass yeast culturing and high quality purification system are needed for structural analysis like as MS and NMR analysis. This kind of experimental approach also needed the multi combination constructions of yeast expression vector, which will take some time to construct. We alternatively chose the study for the elucidation of ACTs and UGTs functions in cucurbitacin synthesis pathway. We could conduct ACTs and UGT enzymatic studies using major cucurbitacins, which are commercially available, as substrats. Unfortunately, only one UGT (UGT74F2) soluble protein was successfully purified out of seven of UGTs in the SI table 2, and three of ACTs soluble protein were purified in pET recombinant protein expression system for in vitro enzyme activity assay. We conduct our study using these four gene and purchased cucurbitacins. And we showed the CYP450s in SI fig. 1 and table 2 to offer information to whom might have interest. We added statement why we used just these four genes in text (line 94–104). We provided fold change in the SI fig. 1 for better comparison (SI line 190–192).

12) Line 67: What is UGT73AM3?

Response: We added introduction about UGT73AM3 in text (line 109–111).

13) Lines 79–80: All NMR data should be presented such that each spectrum can be evaluated by the reader. For example, in all NMR-related Figures, correct axes labels are missing and numbers are very difficult to read, signals cannot be identified as the spectra are plotted too small, peaks are not labeled with the respective atom, the basis for chemical shift assignments is not clear (are there previous studies?), and color-codes are not explained. Especially the assignment procedure has to be presented such that everyone can follow it.

Response: As you mentioned, we corrected NMR spectrum, font size of axes label, size of each

signal, and meaning of each signal, to read easy in supplementary information (SI Figs. 2 to 11).

14) Line 84: How do the authors know that C16 is acetylated by ACT3? Has this been shown in previous studies (no citation is given here)? If not, I strongly suggest showing this by NMR.

Response: *Actually, it can be confirmed using proton NMR spectrum. Because when acetyl moieties connected with some functional group, proton of that functional group is moved to down magnetic field by esterification effects. And also it can be confirmed using HMBC spectrum. According to these rule, we ensured that acetyl groups are connected at C-16. And to read easy, we described detail information of identify each structure in supplementary data (SI line 90–91, 109–111 and 126–129) and add key correlations of HMBC data (SI Figs. 3b, 5b, 7b, 9b, 11b).*

15) Line 90: What is the evidence for the sugar moiety playing a major role in catalytic efficiency?

Response: *Because the only difference between CuE and CuE–Glu was sugar moiety, we thought that the differences of k_{cat} , k_{cat}/K_m and V_{max} value were derived from sugar moiety.*

16) Line 91: From which source do these leaves originate? This is not clear from the Methods section.

Response: *We added information about the source used in cucurbitacins composition analysis in main Methods section (line 260–261 and 263–265).*

17) Line 102: Based on the background information provided by the authors, I was very surprised that ACT2 does not show any enzymatic activity. What is the reason for this? Is this reproducible? The authors should at least include a discussion about it; I miss (possible) explanations for this observation.

Response: *This comment overlap with 23) comment. Please refer to “the 23) comment response” below.*

18) Line 102–120: I was even more surprised that ACT1 acts both as acetylase and deacetylase. It does not become clear how this is possible. Is this observed for other acetyltransferases as well? I strongly suggest that further analyses are included towards establishing a mechanistic explanation of this finding. Or is there strong evidence in precedent literature? A possible starting point could be to investigate whether multiple active centers in ACT1 exist. As long as no mechanism is provided, this conclusion is not convincing.

Response: *We also surprised about this results. It is hard to explain the mechanism of this kind of enzymatic activity. Although we tried to find the precedent literature, we could not find proper reference. We think that further study like as protein x-ray crystallography and enzymatic assay of partially mutated multi ACT1 are need to reveal mechanism. It might be take long time to conduct this kind of research. So, we tone down the statement of line 102–120 (line 162–164). Since we are unable to explain the mechanism of bifunctional enzymatic property of ACT1, we suggest that it will be interesting to determine the mechanism (line 243–248).*

19) Lines 121–129: An explanation why UGT74F2 glycosylates CuE, but not CuB, CuD and CuI is missing.

Response: *We could not explain the mechanism why UGT74F2 glycosylated CuE only. This type*

of substrate preference also confirmed previous studies of UGT7AM3 (Zhong et al, *Molecular plants* 1000–1003, 2017) referenced in next paragraph in text (line 173–174).

20) Line 136–138: I don't understand why only 16-O-acetyl CuE was detected in wounded leaves and not any of the other derivatives, especially as expression of all three ACTs seems to be increased (Fig 3c). This doesn't make sense to me.

Response: We think that other acetylated cucurbitacins (16-O-acetyl CuB, 16-O-acetyl CuD, 16-O-acetyl CuI) by ACT3 were not detectable clearly or not show liable changes because initial contents of CuB, CuD and CuI in watermelon leaves were too low to be detectable in our HPLC system. So, we showed the results of up-regulated 16-O-acetyl CuE content as a representative results. We also added statement about why the other acetylated cucurbitacins were not shown in this study (line 248–253).

21) Lines 149–158: I appreciate that the authors provide quite some background for the in vivo experiment. Some careful editing regarding language can improve this section further.

Response: Thanks. We edited this section (line 194–204).

22) Lines 159–167: Why do the authors choose *Drosophila* as a model system (why is it suitable for this experiment)? Why does 16-O-CuE not enhance neuronal activity in *Drosophila*, even though it was found to be increased by ACT3 in wounded leaves (Fig 3a–d)? Also, the authors conclude that 16-O-acetyl CuB increases neuronal activity (Fig 3e,f and lines 159–161), but they stated that only 16-O-acetyl CuE is found in wounded leaves (lines 136–138). This suggests that the 16-O-CuE stimulation is not relevant. I further think the authors should perform the tip recordings at various 16-O-acetyl CuB concentrations to validate the significance of the observed neuronal stimulation (I find only data for 0.01 mM). What are physiologically relevant concentrations? In addition, the authors should show more examples of recordings as in Fig 3f, for example in the SI, and present them with a longer trajectory over time. The average frequency plot is based on numbers per sec, but only a shorter time window, and only one trajectory per compound is shown in Fig 3f. What is the evidence that the observed increase in neuronal activity is really due to taste? Is it not conceivable, in particular given the small differences observed (Fig 3e), that other effects (e.g. conductivity of the solution) contribute to this increase? Is the receptor in *Drosophila* for these substances known? In summary, I feel that any biological relevance of the newly identified cucurbitacin derivatives is not evidenced by the experiments in the current state of the study.

Response: *Drosophila* is a genetically tractable insect model organism. Actually when we tested two Cucurbitacin forms using wild-type vinegar flies, there is no clear difference in most concentrations. The physiological concentration should be the concentration in the plant leaf (4~25 ug/g CuB or CuE in watermelon leaf, Kim et al, *Hort. Environ. Biotechnol.* 2018 59:557–566). In another study, one gustatory receptor from genetic screening of candidate chemoreceptors was found (Cucurbitacin B Activates Bitter-Sensing Gustatory Receptor Neurons via Gustatory Receptor 33a in *Drosophila melanogaster*. Rimal, Sang, Dhakal, and Lee, *Mol & Cells* in press). The tip recording using bitter compounds such as caffeine, strychnine, and others is typically only active during ~500ms and then the amplitude is quickly dampened by adaptation (Lee Y., Moon SJ, and Montell C. *PNAS* 2010; Weiss LA, Dahanukar A, Kwon JY, Banerjee D, Carlson JR. *Neuron*. 2011). However, we found that CuB might have a little different kinetics. As you see in the new figure SI Fig. 17, the response is relatively persistent (SI line 281–287).

The observed increase in neuronal activity disappeared when we induced apoptosis of bitter-sensing gustatory receptor neurons (GRNs) (Rimal, Sang, Dhakal, and Lee, *Mol & Cells* in

press). Furthermore, it is dependent on *Gr33a*, gustatory receptor. So we conclude that this amplitude is dependent on bitter-sensing GRNs and *Gr33a*. We added this information in the main text (line 204–206 and 209–211).

23) Lines 171–186: I don't quite understand the point of performing these studies. I don't think there is strong experimental evidence in these analyses supporting the hypothesis that ACT acetylation is an evolutionarily conserved mechanism. Also, ACT2 did not show any activity for the tested plant cucurbitacins, but now for ecdysone and cortisol (Fig 4b,c). How can this be explained? I am skeptical that ACT2, being an enzyme from plants, catalyzes acetylation of triterpenoid of animals, but not of plants. Is the missing ACT2 enzymatic activity on the plant cucurbitacins really reproducible? Can the authors provide any further experimental evidence?

Response: We change the statement “ACT acetylation is an evolutionarily conserved mechanism” to “the minor structural differences could affect the acetylation capability of these ACTs”. We could not explain mechanism of ACT2 substrate preference. We conducted additional ACT2 enzymatic activity assay, recently. We attached the additional results showing reproducible of ACT2 enzymatic activity against other plant cucurbitacins. In left column of figures “without enzyme” means the enzymatic reaction containing cucurbitacins (CuB, CuD, CuE and CuI) and no ACT2, and right column of figures are results of enzymatic reaction containing both of cucurbitacins (CuB, CuD, CuE and CuI) and ACT2. We could not see the additional peaks in ACT2 enzymatic reaction products when compare with “without enzyme” reactions.

Sample Name : W/O_CuB
 Sample ID : W/O_CuB
 Data Filename : 200515_6 lod
 Method Filename : CuBE.slm
 Batch Filename : ACT substrate.lcb
 Vial # : 1-1
 Injection Volume : 20 µL
 Date Acquired : 2020-05-15 @ 12:53:42
 Date Processed : 2020-05-15 @ 1:38:43
 Sample Type : Unknown
 Acquired by : System Administrator
 Processed by : System Administrator

<Sample Information>
 Sample Name : ACT2_CuB
 Sample ID : ACT2_CuB
 Data Filename : 200515_1 lod
 Method Filename : CuBE.slm
 Batch Filename : ACT substrate.lcb
 Vial # : 1-1
 Injection Volume : 20 µL
 Date Acquired : 2020-05-14 @ 1:39:03
 Date Processed : 2020-05-14 @ 2:24:04
 Sample Type : Unknown
 Acquired by : System Administrator
 Processed by : System Administrator

24) The legend of Fig 4d is missing.

Response: We added the legend of Fig 4d (line 598–600).

REVIEWERS' COMMENTS:

Reviewer #2 (Remarks to the Author):

The authors have revised their manuscript substantially and addressed all my points. No further revision is required. I congratulate the authors for their contribution to biosynthesis of cucurbitacin and derivatives thereof.